# SARS-CoV-2 Mpro responds to oxidation by forming disulfide and NOS/SONOS bonds

The main protease (Mpro) of SARS-CoV-2 is critical for viral function and a key drug target. Mpro is only active when reduced; turnover ceases upon oxidation but is restored by re-reduction. This suggests the system has evolved to survive periods in an oxidative environment, but the mechanism of this protection has not been confirmed. Here, we report a crystal structure of oxidized Mpro showing a disulfide bond between the active site cysteine, C145, and a distal cysteine, C117. Previous work proposed this disulfide provides the mechanism of protection from irreversible oxidation. Mpro forms an obligate homodimer, and the C117-C145 structure shows disruption of interactions bridging the dimer interface, implying a correlation between oxidation and dimerization. We confirm dimer stability is weakened in solution upon oxidation. Finally, we observe the protein's crystallization behavior is linked to its redox state. Oxidized Mpro spontaneously forms a distinct, more loosely packed lattice. Seeding with crystals of this lattice yields a structure with an oxidation pattern incorporating one cysteine-lysine-cysteine (SONOS) and two lysine-cysteine (NOS) bridges. These structures further our understanding of the oxidative regulation of Mpro and the crystallization conditions necessary to study this structurally.

During the COVID-19 pandemic, the SARS-CoV-2 main protease (Mpro, nsp5 or 3CLpro) emerged as a key antiviral target and focus of intense study[1–3]. Mpro plays a central role in the SARS-CoV-2 replication cycle, as the viral genome codes for polyproteins that must be cleaved into individual protein units to support viral function. Mpro processes at least 11 known sites along polyproteins 1a and 1ab, including its own N- and C-termini[1], and is therefore essential for viral replication. This key role in replication, along with the historical success of viral protease inhibitors, the lack of any similar human protein, and prior work on SARS-CoV-1 Mpro, has made SARS-CoV-2 Mpro the target of several drug discovery programs. These efforts have already yielded an FDA approved molecule, nirmatrelvir[4]. Given the persistence of the COVID-19 virus and the possible emergence of future pathogenic coronaviruses, it is imperative we develop a deeper understanding of Mpro and its role in viral function.

Mpro's activity is regulated by multiple mechanisms, though we have a poor understanding of how these support viral fitness. Most prominently, at sufficiently high concentrations, the enzyme forms a homodimer. Dimerization enhances the catalytic rate, effectively turning Mpro from an inactive form into an active one[5]. Structural work suggests this concentration-dependent regulation is not an evolutionary accident. Specifically, Mpro adopts a chymotrypsin-like fold but has a distinct dimerization domain at its C-terminus that many other chymotrypsin-like enzymes lack[6]. Studies of the truncated enzyme without this domain, as well as of the domain in isolation, have demonstrated it is both necessary and sufficient for dimer formation[7]. This suggests that this dimerization domain enables regulation of Mpro's catalytic rate based on the concentration of free enzyme in the cell.

In addition to regulation via dimerization, Mpro has been shown to be sensitive to the local redox environment. Including the active site cysteine, the protein sequence contains 12 cysteine residues (~4% of total), an unusually high number[8]. Under mildly reductive conditions all cysteines are reduced, and the protein's catalytic rate is maximized, suggesting this is the active form of the enzyme found in a cellular context[9]. Upon oxidation, a remarkable and growing number of modifications have been reported by both structural and mass spectrometry

✉ e-mail: thomas.lane@desy.de

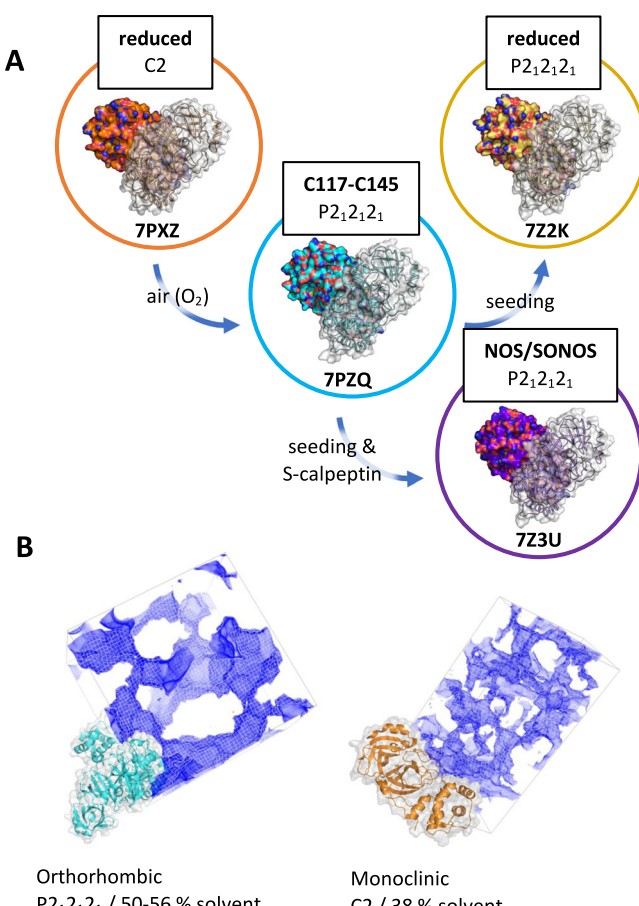

**A**

reduced
C2

**7PXZ**

air (O₂)

C117-C145
P2₁2₁2₁

**7PZQ**

seeding &
S-calpeptin

reduced
P2₁2₁2₁

**7Z2K**

seeding

NOS/SONOS
P2₁2₁2₁

**7Z3U**

**B**

Orthorhombic
P2₁2₁2₁ / 50-56 % solvent

Monoclinic
C2 / 38 % solvent

**Fig. 1 | The redox state of M^pro is linked to its crystallization behavior. A** Reduced protein under our crystallization conditions, containing TCEP, results in monoclinic (C2) protein crystals (PDB ID: 7PXZ). In a separate batch crystallization experiment, after oxidation by air and in the absence of reducing agent, the same protein spontaneously forms crystals with an orthorhombic (P2₁2₁2₁) lattice and exhibits a disulfide link between C117 and C145 (PDB ID: 7PZQ). By producing seed crystals from this oxidized protein, however, we were able to obtain two further structures in the same orthorhombic lattice: first, using reduced protein, a structure with reduced C117/C145 (PDB ID: 7Z2K), and second, using the same reduced protein but with the addition of a sulfonated calpeptin ligand, a structure exhibiting NOS and SONOS crosslinks (PDB ID: 7Z3U, which is published in ref. 20). **B** Visualization of the orthorhombic and monoclinic lattices, with the solvent content highlighted by *map-channels*[47]. The packing and crystal contact patterns are substantially altered, with the orthorhombic lattice exhibiting significantly larger solvent channels and an overall higher solvent content.

studies, including glutathionylated C300[10], a peroxy-C145, an N-ethylmaleimide modified C145 and C156[11], a SONOS bridge between C22, C44, and K61[12,13], and a disulfide link between C117 and C145[9,14].

Given the complexity of viral replication in human hosts, the prevalence or role of these modifications in the viral replication cycle has remained unclear. Oxidative stress in the cell has been shown to regulate the function of other viruses[15], most notably HIV[16–18], and early in the pandemic oxidative stress was hypothesized to play a central role in COVID-19 pathogenesis[19]. It has even been speculated that robustness to oxidative environments might enable corona or other viruses to survive in bat hosts, which are known to exhibit unusual oxidative cellular conditions[10].

Among the known oxidative modifications of M^pro, the C117-C145 disulfide modification particularly is notable. Funk and colleagues recently performed a systematic study of the behavior of M^pro under oxidative conditions and highlighted this modification as uniquely functional[9]. They produced single-point cysteine-to-serine mutants for

each cysteine in M^pro. Of all these mutants, they found C117S was the only mutant that did not recover activity after being exposed to H₂O₂ and then re-reduced with DTT. This suggests C117 may have a special role in protecting the active site C145 from oxidative damage. Supporting this, Tran et al. recently reported the structure of an oxidized mutant of M^pro, the H163A variant, that contained a disulfide bond linking C117 to C145[14]. The H163A modification inactivates the enzyme, even under reducing conditions. Removal of the H163 sidechain disrupts a π-stacking interaction with F140, which is part of the loop that forms the transition state-stabilizing oxyanion hole. Disruption of the oxyanion hole was observed in structures of both the oxidized (C117-C145) and reduced (no disulfide) enzyme, explaining the loss in activity. However, it was unclear how the C117-C145 disulfide in the H163A variant related to the behavior of the wild type.

To address these questions, we determined the structure of M^pro with the C117-C145 disulfide modification under mildly oxidizing conditions, providing a structural understanding of how this disulfide can protect the enzyme from irreversible oxidation. We find that oxidized protein only crystallizes in a more loosely packed, orthorhombic lattice, whereas the reduced protein forms a monoclinic lattice under the same crystallization conditions. Seeding with these orthorhombic crystals enabled us to crystallize M^pro exhibiting a set of NOS and SONOS oxidative modifications.

## Results

### An orthorhombic lattice is flexible enough to produce crystals with oxidative modifications

By delivering streams of microcrystals into the x-ray focus of the SPB/SFX instrument of the European XFEL, we obtained diffraction data yielding two crystal structures of M^pro, one active/reduced structure and one inactive/oxidized structure (supplementary Table S1). The crystals merged to determine these two structures were grown in two separate batch crystallization experiments. In the first, the resulting structure was fully reduced due to the presence of 1 mM TCEP during purification. In the second experiment, TCEP was omitted, and M^pro exposed to air over time spontaneously crystallized into a different space group, despite being crystallized under otherwise the same conditions (buffer, temperature, concentrations). Specifically, our reduced-M^pro crystals formed a monoclinic lattice with C2 symmetry. These crystals contain the native homodimer, with a single protomer in the asymmetric unit and the dimer completed by crystallographic symmetry. Protein subjected to oxidation by exposure to air exhibits a covalent disulfide bond between C117 and C145 and forms crystals in space group P2₁2₁2₁, with the asymmetric unit consisting of the entire homodimer (protomer A-to-B all-atom RMSD: 0.96 Å). The orthorhombic lattice exhibits a looser overall packing and higher solvent content (Fig. 1, supplementary Table S1). Both datasets were collected at room temperature.

While crystallization conditions for the oxidized and reduced crystals are the same, the lattices and crystal morphologies obtained differ (Fig. 1, supplementary Fig. S1). This makes a direct comparison of reduced and oxidized structures challenging, as we could not control for differences due to oxidation state vs. crystal packing. Therefore, we attempted to obtain a reduced structure in the orthorhombic lattice seen in our oxidized crystals. By seeding reduced protein with crystals of the oxidized form, we were successfully able to generate crystals of reduced protein in the orthorhombic lattice (Fig. 1). As our XFEL beamtime had concluded by this time, data for these crystals were collected under cryogenic conditions at PETRA-III beamline P11 (supplementary Table S1). The cryogenic conditions cause a contraction of the lattice and reduction of the solvent content by 4–5% as compared to room temperature collection (supplementary Table S1). The molecular structure of the enzyme in the reduced state is similar in both the monoclinic (XFEL/RT) and the orthorhombic (synchrotron/100 K) lattices (all-atom RMSD: 1.56 Å). Both structures are used here as a basis of comparison to elucidate changes due to oxidation.

Finally, in conjunction with our ongoing work to develop M[pro] inhibitors, we employed our oxidized orthorhombic seeds in a co-crystallization experiment with M[pro] bound to a sulfonated calpeptin derivative. The resulting structure was reported in a paper describing the ligand binding pose[20]. Unexpectedly, the same structure exhibits a rich pattern of oxidative modifications, which we discuss here. Protomer A contains a SONOS bridge involving C22, C44, and K61, whereas protomer B shows only a NOS bridge involving C22 and K61 at the same site. Both modifications are consistent with previous reports[12,13]. In addition, 2mF$_o$-DF$_c$ maps unambiguously show a NOS bridge between K102 and C156 in protomer B, not previously described in the literature, and suggest partial occupancy of the same modification in protomer A.

### Disulfide formation in M[pro] precludes catalysis and disrupts the dimer interface

By oxidizing M[pro] via air exposure, we obtained structures with a disulfide bond between C117 and C145. To understand the structural changes that occur upon formation of the C117-C145 disulfide bond (Fig. 2), we determined two reduced reference structures. The first was obtained at room temperature with XFEL radiation, identical to the data collection conditions of our C117-C145 structure, but crystallized in a different space group (C2). The second reference structure is in the same space group as the C117-C145 structure (P2$_1$2$_1$2$_1$), following seeding with crushed oxidized crystals, and was obtained at 100 K.

In the reduced form, the catalytically active cysteine C145 sits on a loop in the active site pocket, while C117 forms part of a β-hairpin about 8 Å away (C$_\alpha$-to-C$_\alpha$). Oxidative crosslinking of these residues relocates both to a location approximately in the middle of their reduced positions (5.1 Å C$_\alpha$-to-C$_\alpha$). This disrupts the β-hairpin motif containing C117 and displaces the conserved N28, which in the reduced structure sits between C145 and C117 but in the oxidized structure undergoes a rotamer shift to make space for the disulfide bridge (Fig. 2). This residue was identified as essential for dimerization and enzymatic activity in SARS-CoV-1 M[pro] [21]. The rotameric change of N28 was predicted by MD simulations performed by Funk et al. and

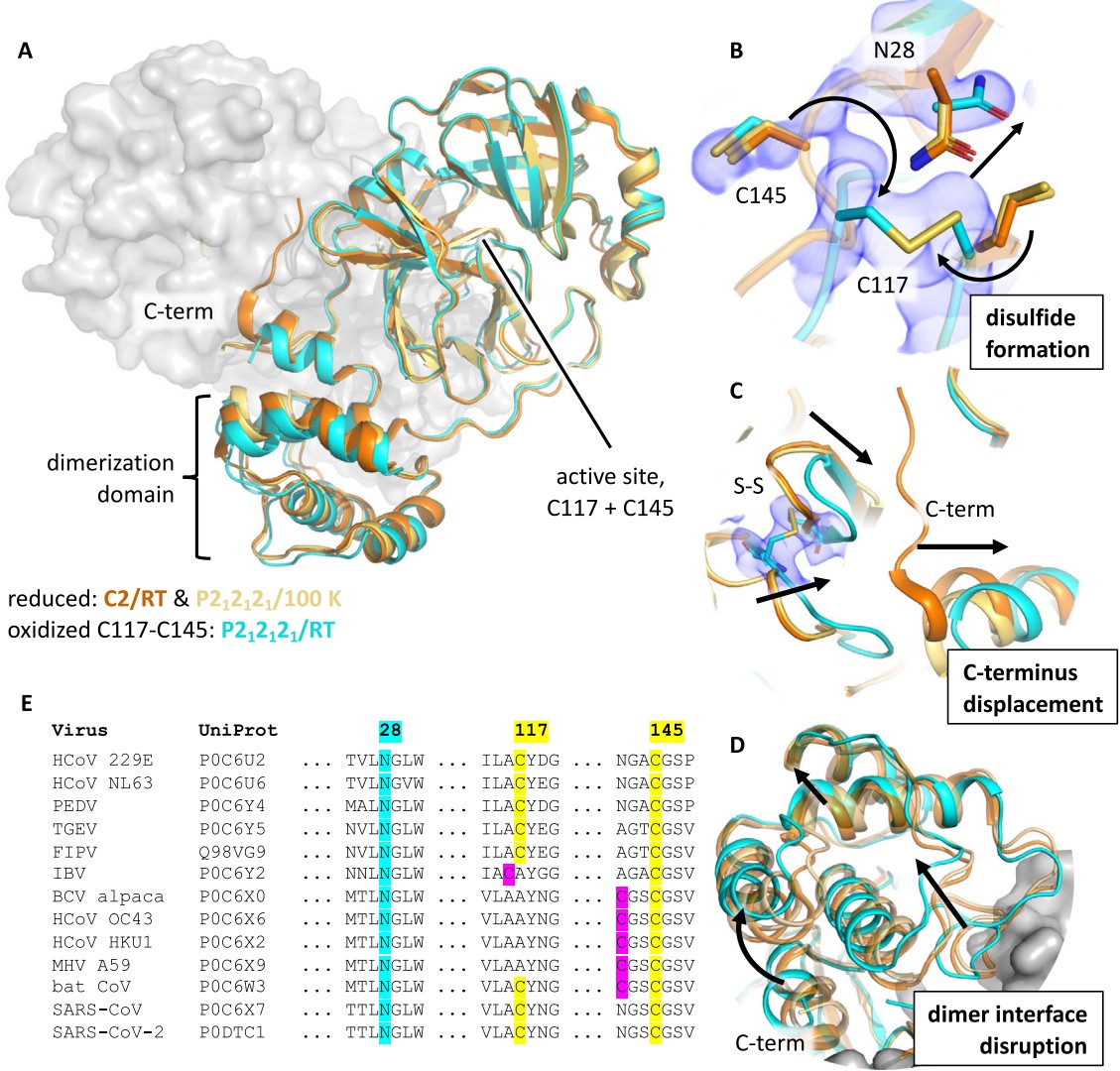

reduced: **C2/RT** & **P2$_1$2$_1$2$_1$/100 K**
oxidized C117-C145: **P2$_1$2$_1$2$_1$/RT**

**Fig. 2 | Long-range structural changes correlated with C117-C145 disulfide formation disrupt the dimer interface. A** aligned overlay of reduced (orange: monoclinic/room temperature, PDB ID: 7PXZ; yellow: orthorhombic/100 K, PDB ID: 7Z2K) and oxidized (cyan: orthorhombic/room temperature, PDB ID: 7PZQ) structures, with one monomer of the M[pro] dimer shown as surface. Oxidation of the active site cysteine, C145, results in **B** disulfide bridge formation with C117 and displacement of N28 (density: oxidized 2mF$_o$-DF$_c$ at 1 RMSD). Colocalization of C117 and C145 requires **C** displacement of C-terminal residues 301–306 from the dimer interface and is correlated with **D** a shift of the dimerization domain and disruption of the stabilizing interactions between the two protomers. **E** MSA showing N28, C117, and C145 are conserved across related coronaviruses. N28 and C145 are absolutely conserved in the set studied. C117 is partially conserved, but where it is not, another cysteine is present in either position 116 or 142 (magenta) that could conceivably fulfill the same role.

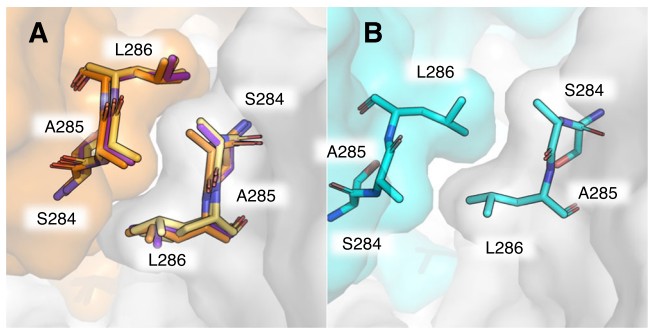

**Fig. 3 | Disruption of the S284/A285/L286 dimerization interface in the disulfide-containing structure.** Shown is the region where the loop containing S284, A285, and L286 forms a hydrophobic zipper with the same residues on the opposite dimer-forming protomer. All four structures reported are drawn, **A** reduced/C2 (orange, PDB ID: 7PXZ), reduced/P2₁2₁2₁ (yellow, PDB ID: 7Z2K), NOS/SONOS (purple, PDB ID: 7Z3U), and **B** C117-C145 (teal, PDB ID: 7PZQ). Only the C117-C145 structure shows a disruption of this dimer interface. For these three residues, the buried surface area decreases from ~110 Å$^2$ for the structures in **A** to 87 and 85 Å$^2$ for molecules A and B of the C117-C145 structure shown in **B**, respectively. Structures were aligned by minimizing all heavy atom RMSD prior to visualization. Surfaces are the solvent-accessible (Connolly) surface computed with the PyMOL Molecular Graphics System (2.0, Schrödinger LLC)[48].

is confirmed by our structures[9]. Moreover, the position of the loop containing the active cysteine, spanning S139–S147, rearranges upon disulfide formation (supplementary Fig. S2). Finally, disulfide formation partially buries the active site cysteine, which has a solvent-exposed area of 24.3 Å$^2$ in the reduced structure but 17.5 Å$^2$ and 13.9 Å$^2$ for the two molecules in the asymmetric unit, A and B respectively, in the oxidized structure.

In protomer A of our oxidized structure, C145 shows the residual population at the same position it occupies in the reduced structure. The electron density of this residual population was sufficient to model an alternative conformation, resulting in a refined structure with 55% occupancy of the disulfide conformer and 45% population of the reduced conformation. In protomer B, the reduced conformation is insufficiently populated to generate a confident model, and our structure contains a fully occupied disulfide. No evidence of other oxidative modifications was observed in our electron density maps.

The structural rearrangements required to bring C117 and C145 together require a series of long-range structural changes that disrupt the dimer interface (Fig. 2). In the reduced state, the C-termini form part of the dimer interface adjacent to the β-hairpin containing C117, but upon crosslinking of C117 and C145, C-terminal residues 301–306 become disordered due to the loop rearrangements necessary to bring the two cysteines together.

This ejection of the C-termini from the dimer interface is accompanied by a shift in the entire dimerization domain, which contains contacts that bridge the two protomers. As a result, the entire dimer interface is less well-packed in the oxidized structure as compared to its reduced counterpart. The surface area that forms the dimer interface is estimated to be 1301 Å$^2$ and 1283 Å$^2$ for protomers A and B, respectively, in the orthorhombic reduced structure. This interfacial area decreases to 1198 Å$^2$ and 1259 Å$^2$ in the disulfide-containing structure. Notably, the loop formed by residues S284, A285, and L286, which packs tightly with the same residues on the symmetric protomer in the reduced state, is disrupted in the oxidized structure (Fig. 3). In the reduced structure, this loop forms a tight zipper-like packing interface with the opposite protomer, but in the oxidized structure this zipper is out of register and does not form a tight interface (Fig. 3). This disruption of the dimer interface suggests M$^{pro}$'s dimer affinity is weakened upon oxidation, as recently reported by Funk et al.[9].

## Analytical size exclusion chromatography confirms weakened dimer affinity

To test if the disruption of the dimer interface observed in our C117-C145 structure translates into a reduction of the dimerization affinity, we performed analytical size exclusion chromatography (Fig. 4). Freshly purified protein, exposed to air for only a few hours, exhibited a dimerization dissociation constant (K$_D$) of 2.3 μM. Addition of 1 mM TCEP in the running buffer produced no significant change, with an estimated K$_D$ of 2.1 μM. However, incubating the protein in 1 mM hydrogen peroxide prior to injection significantly decreased the dimer stability, resulting in an order of magnitude increase of the measured K$_D$ to 19 μM. Further, upon exposure to peroxide, the observed monomer peak elutes notably earlier, at approximately 0.49 column volumes vs. 0.47, suggesting oxidation has produced a less compact monomer species. Our results agree with analytical ultracentrifugation performed by Zhang et al. (K$_D$-2.5 μM, reduced)[2] and the SAXS measurements of Silvestrini et al. (K$_D$-7 μM, reduced)[22]. They are qualitatively consistent with the analytical centrifugation measurements performed by Funk et al.[9], who determined absolute K$_D$ values that are a factor of 10 smaller, but with the same relative order of magnitude decrease upon oxidation.

## Observation of NOS and SONOS modifications upon co-crystallization with a sulfonated calpeptin ligand

During our ongoing structural studies of M$^{pro}$ ligands, we sought to obtain a structure of M$^{pro}$ bound to a ligand of interest, a sulfonated calpeptin derivative that reacts covalently with the catalytic cysteine in M$^{pro}$. Co-crystallization attempts of this ligand with reduced, monoclinic seeds failed to yield a high-resolution structure in our hands, instead forming small clusters of crystals that diffracted to low resolution (-5 Å) and could not be indexed. We hypothesized that the looser packing of the orthorhombic lattice provided by our oxidized seeds might better accommodate structural rearrangements caused by ligand binding. Subsequently, we attempted crystallization with our orthorhombic, oxidized seeds and obtained a high-resolution structure clearly showing bound ligand, which we refined against data up to 1.72 Å. Ligand density consistent with full occupancy was present in the active site of both monomers. Unexpectedly, however, the structure shows multiple NOS and SONOS modifications (Fig. 5).

The NOS and SONOS modifications exhibit a distinct asymmetry between the monomers that form the dimer in the asymmetric unit. Protomer A shows a SONOS linkage between C22, K61, and C44 (Fig. 5), which has been previously reported[12,13]. This modification results in a shift of the α-helix between E55 and K61 and disrupts the position of a loop between C44 and Y54, which is ordered in the reduced structure but becomes disordered upon SONOS formation. This disorder may be in part because the shifted loop can no longer form a backbone H-bonding contact between M49 and Q189, the latter of which sits on a flexible domain-connecting loop consisting of the residues V186 to G195.

In contrast, protomer B more closely resembles the reduced structure. It exhibits a NOS bridge between C22 and K61 (Fig. 5). The effect of this modification is less dramatic, with C22 and K61 separated by 7.2 Å in the reduced structure (C$_\alpha$-to-C$_\alpha$) but only 7.4 Å with the NOS bridge present.

Finally, both protomers A and B show evidence for a NOS bridge between K102 and C156 that connects neighboring β-sheets (Fig. 5, see supplementary Fig. S3 for supporting OMIT and isomorphous difference maps). This modification is clear in the density for protomer B, while the density in protomer A is ambiguous but consistent with a NOS bridge at low occupancy. The NOS modification at this site induces essentially no deviation from the reduced structure in the same space group, where K102 and C156 are in close proximity.

## Discussion

M$^{pro}$ appears to exhibit an unusually rich set of oxidation modifications, which have been revealed by structural and biochemical methods.

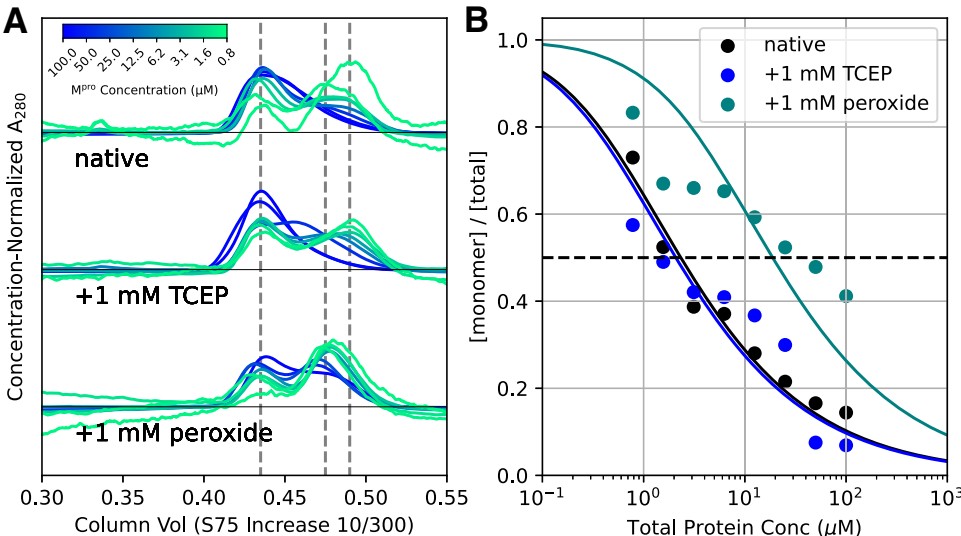

**Fig. 4 | Size exclusion measurements show a shift in monomer-dimer equilibrium upon oxidation. A** M$^{pro}$ dimerization was analyzed under three conditions: first, immediately after purification, without oxidizing or reducing agent (native, top); second, in the presence of 1 mM TCEP in the running buffer (middle), or after the protein was incubated with 1 mM peroxide for 5 h (bottom). Protein was logarithmically diluted between 100 μM and 780 nM and assayed by size exclusion. SEC traces are normalized by protein concentration so that they appear on the same scale; lower concentrations therefore exhibit visible baseline drift and noise. Distinct monomer and dimer peaks are seen for each sample/concentration combination, eluting at approximately 0.48 and 0.43 column volumes, respectively. The concentration of these species was estimated by the peak height at the positions indicated by vertical dashed lines. The peroxide-exposed monomer (middle dashed line) elutes earlier than the monomer peaks for native and TCEP-exposed protein (rightmost dashed line), suggesting this species is modified and may be structurally less compact than fully reduced monomer. **B** The monomer and dimer concentrations from **A** were fit to a two-state equilibrium model, $2M \rightleftharpoons D$ (solid lines), with a single parameter, the $K_D$. The determined $K_D$ parameters show that oxidation of M$^{pro}$ by 1 mM peroxide (teal, $K_D$ 19 ± 14.8 μM) significantly weakens the dimer interface as compared to reduced protein (blue, $K_D$ of 2.1 ± 0.50 μM, 1 mM TCEP in the running buffer) or freshly purified "native" protein in the absence of exogenous reducing or oxidizing agents (black, $K_D$ 2.3 ± 0.63 μM). Errors reported are 95% confidence intervals assuming a Gaussian error model. The x-axis reports the total concentration of single M$^{pro}$ protein chains. The y-axis reports the fraction of monomeric chains as determined from the traces in (**A**).

While a response to oxidative stress has been implicated in virus biology in general, the possible physiological relevance of each of the observed oxidized states of M$^{pro}$ remains a topic of ongoing investigation.

Our structure of C117-C145 modified M$^{pro}$ provides a mechanistic model for several key observations regarding M$^{pro}$'s behavior upon change of redox state. Most notably, our structures provide a simple explanation as to why M$^{pro}$'s dimer affinity decreases by about an order of magnitude upon oxidation (Fig. 4)[9]. Our structure further confirms a key role of N28, which rotates to allow space for the C117-C145 disulfide bridge. N28 is highly conserved (Fig. 1), suggesting asparagine at this position is essential for viral fitness[21]. We assume that the small volume and hydrophilic nature of the carboxamide sidechain facilitates this conformational change, enabling M$^{pro}$'s ability to toggle between reduced and oxidized states.

While we have been able to grow large single crystals from air-oxidized protein in the orthorhombic space group (supplementary Fig. S1) and have studied these with synchrotron radiation, we have not observed clear C117-C145 disulfide density in those studies. In contrast, the diffraction of XFEL light from microcrystals clearly shows the C117-C145 disulfide. We considered the hypothesis that XFEL radiation may have allowed us to observe this modification via "radiation damage-free" data collection, as the x-ray exposure (<100 fs) is much more rapid than the nuclear motions required for the two cysteine sidechains to adopt significantly different positions following x-ray induced reduction[23–25]. In this model, structures obtained at the synchrotron would appear fully reduced, because during data collection at the synchrotron, x-ray-induced cleavage of the disulfide would reverse the structural changes that accompany C117-C145 crosslinking. This would require an implausibly large rearrangement of the structure under cryogenic conditions. The success of Tran et al. in capturing this disulfide with synchrotron radiation in the H163A variant[14] is further evidence against this model. Their ability to obtain a structure

clearly showing the C117-C145 disulfide suggests that this modification is not uniquely sensitive to reduction by X-rays.

Alternatively, it may be that the large crystals containing significant C117-C145 population are less well-ordered than their microcrystalline counterparts, and XFEL radiation allowed us to study such microcrystals. Our data only show that XFEL radiation is sufficient to observe the C117-C145 disulfide in wild-type M$^{pro}$, but we cannot conclude that such radiation is necessary to preserve the C117-C145 modification.

By studying the H163A M$^{pro}$ mutant, Tran and colleagues were able to crystallographically observe the same C117-C145 disulfide bond we found in wild-type M$^{pro}$[14]. In that work, they observed that the loss of the imidazole sidechain in H163 removed a favorable π-stacking interaction with F140, resulting in displacement of a mobile loop containing F140 (supplementary Fig. S2). Further, this loop motion displaces the N and C termini, resulting in dimer interface disruption. Importantly, the loop containing F140 forms the oxyanion hole, which stabilizes the transition state during proteolysis. In our disulfide-containing structure, the F140 loop does not reach the same extreme position as observed by Tran et al. but is disrupted from its conformation in the corresponding reduced structures (supplementary Fig. S2). Our work therefore supports the hypothesis of Tran et al., that C117-C145 can form in a similar fashion in both the wild type and H163A enzymes but demonstrates that the extensive remodeling of the F140 loop in the H163A structure is particular to that variant.

Complementing these structures of the C117-C145 disulfide, Funk et al. reported that M$^{pro}$ C117S was the only C-to-S mutant that failed to recover activity after exposure to oxidative conditions followed by reduction[9]. Our structures illustrate how, upon oxidation, the catalytic C145 moves from a solvent-exposed conformation to a buried, disulfide conformation. Our structure, alongside these previous findings and the conserved nature of cysteines at positions 117 and 145,

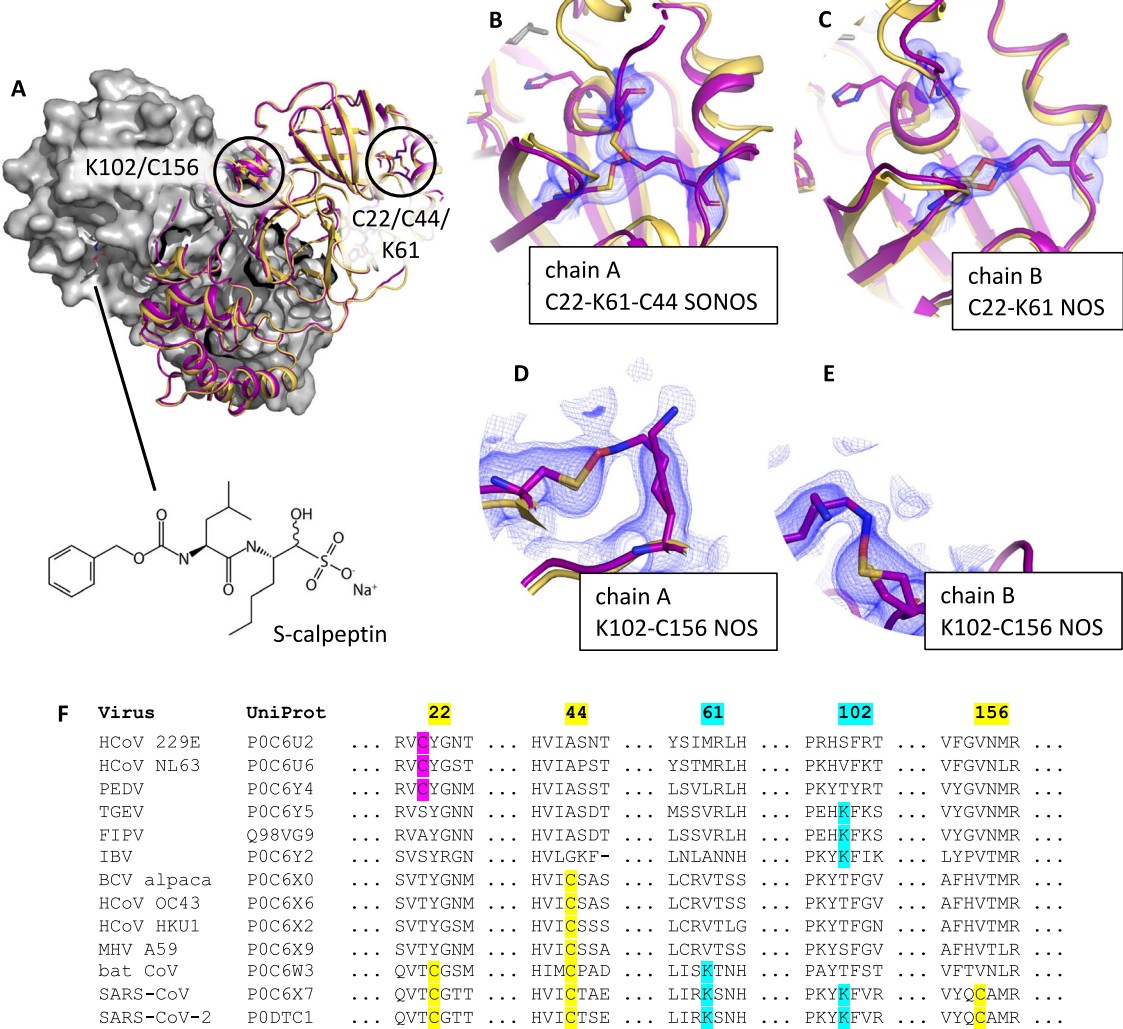

**Fig. 5 | NOS/SONOS crosslinking observed upon co-crystallization with sulfonated calpeptin in the orthorhombic space group.** Oxidative NOS and SONOS bridges are seen at four sites in the **A** two dimeric protomers that form both the asymmetric and biological unit (all panels, purple: oxidized NOS/SONOS structure in P2₁2₁2₁, PDB ID: 7Z3U, published in citation 21; yellow: reduced reference in P2₁2₁2₁ PDB ID: 7Z2K). **B** Protomer A exhibits a SONOS linkage between C22, K61, and C44 that distorts the structure from the reduced form. In contrast, **C** protomer B shows only a NOS linkage between C22 and K61 at this site, with the overall structure differing little from the reduced reference. **D** The electron density on protomer A between K102 and C156 is ambiguous, consistent with but not conclusively showing a NOS linkage at partial occupancy. In contrast, **E** the density at the same site on protomer B clearly shows a NOS bond at partial occupancy. **F** None of the residues participating in these linkages are strongly conserved besides C44. Densities shown as blue volumes are the 2mF$_o$-DF$_c$ map at 1 RMSD. The same 2mF$_o$-DF$_c$ map at 0.5 RMSD is overlaid as a light mesh for the K102-C156 NOS figures, to show partial occupancies more clearly.

implicate this modification in a regulatory response to an oxidative environment. Our results support the idea that the C117-C145 disulfide provides a protective mechanism against oxidative damage by making harsher, irreversible oxidation of the catalytic C145 to sulfinic and sulfonic acids impossible[26].

Seeding with crystals containing the C117-C145 disulfide enables kinetic control over the crystallization lattice, allowing us to obtain a ligand-bound structure that shows NOS and SONOS bridges. We considered the hypothesis that ligand binding might facilitate these modifications. Yang and colleagues, however, recently presented nine M$^{pro}$ structures exhibiting the C22-K61-C44 SONOS bond. Five contain bound inhibitors but four show no ligand of interest[13], demonstrating that ligand binding is not necessary to observe SONOS modifications in M$^{pro}$ crystals. Further, the NOS bond at K102-C156, which has not been reported previously, is far from the active site and seems unlikely to be influenced by the binding of a ligand in that pocket. As no oxidizing agents were added to the crystallization experiment, we attribute NOS/SONOS formation to molecular oxygen introduced by air exposure.

The frequency and diversity of NOS/SONOS modifications observed in M$^{pro}$ suggest these crosslinks may have a functional role in regulating the enzyme's function in oxidative environments.

The fact that M$^{pro}$ is the premier target for anti-COVID-19 small molecule therapeutics warrants further investigation into the possible impact of these oxidative modifications on viral fitness. While our structures provide mechanistic insight into the regulation of the enzyme by oxidation and dimerization, we must now understand how regulation based on oxidative stress or protein concentration impacts the viral lifecycle in vivo. Such work will provide deeper insight into viral biology and – hopefully – open new opportunities to disrupt that biology with life-preserving medicines.

## Methods
### Protein production and purification
The gene encoding M$^{pro}$ was overexpressed in *E. coli* and purified for subsequent crystallization using previously published protocols and plasmid constructs[2]. Specifically, cell pellets containing recombinant

protein were lysed in 20 mM TRIS buffer, pH 7.8, supplemented with 150 mM NaCl and 10 mM imidazole using a homogenizer. After removal of insoluble cell matter by ultracentrifugation, a nickel NTA column was used to purify the M[pro]-histidine-tag fusion protein. Following imidazole elution, the protein buffer was changed to 20 mM TRIS, pH 7.8, 150 mM NaCl, 1 mM TCEP using a PD10 column, and the histidine tag was cleaved by 3C protease overnight. Subsequently, the histidine tag and the 3C protease were removed using a nickel NTA column. For the reduced form of M[pro] a final size exclusion chromatography was performed with an S200 Superdex column using 20 mM TRIS, pH 7.8, 150 mM NaCl, 1 mM TCEP, and 1 mM EDTA, while for the oxidized form TCEP was omitted.

## Crystallization experiments

M[pro] microcrystals were grown using seeded batch crystallization in the XBI laboratories of the European XFEL[27]. Seed crystals were grown from protein purified in the presence of 0.5 mM TCEP, using a sitting drop geometry by combining 250 nL M[pro] protein solution (6.25 mg/ml) and 250 nL precipitant (25% PEG1500, 0.1 M MIB buffer pH 7.5, 5% DMSO), as reported previously[28]. A seed stock was produced by adding the resulting M[pro] crystals to a reaction tube containing a glass bead (Beads-for-Seeds, Jena Bioscience) and vortexing periodically for 5 s with subsequent incubation at room temperature. For the microcrystal batch crystallization, 250 μL glass seed beads were added to a 1.5 mL reaction tube, which was then filled with 900 μL precipitant solution (25% PEG1500, 0.1 M MIB buffer pH 7.5, 5% DMSO) mixed with 100 μL seed stock and 100 μL M[pro] protein solution (35 mg/ml). Subsequently, crystals were grown in a shaker at 18 °C at 900 rpm overnight. The reduced and oxidized forms were crystallized separately. Resulting crystals were thin plates with a size ranging from 3–15 μm. Crystal concentration was adjusted by allowing the crystals to settle overnight and removing supernatant accordingly. Final crystal slurry was filtered through a 30 μm mesh gravity filter (Sysmex CellTrics) before injection.

Protein crystals for single-crystal rotation experiments were produced as previously reported[28], using orthorhombic seeds and reduced protein at 6.25 mg/mL. For the ligand free and S-calpeptin containing crystallization experiments, the same reduced protein batch was used. The S-calpeptin compound was dried in the well prior to crystallization mixture addition, yielding a maximum concentration of 5 mM.

## Instrumentation

SFX experiments (7PXZ, 7PZQ) were performed at the SPB/SFX instrument[29] in April 2021 as a part of proposal 2696. The size of the mirror-focused focal spot in the interaction region was estimated to be $4 \times 4\,\mu m^2$ FWHM diameter based on optical imaging of single shots using a 20 μm thick Ce:YAG screen. The x-ray pulse energy was in the range of 1.2–3.5 mJ at 9.3 keV. Diffraction from the sample was measured using an AGIPD[30] of 1 megapixel located 117.7–118.6 mm downstream of the sample interaction region, with the unused direct beam passing through a central hole in the detector to a beam stop further downstream. The resolution at the edge of the AGIPD was 1.8 Å, and 1.6 Å data were obtained by integrating Bragg reflections up to the detector corner. Experiment control was provided by Karabo[31].

We used double-flow focusing nozzles (DFFN) for sample delivery[32,33]. The DFFN had an inner diameter of 75 μm and a liquid jet was established by applying 35 mg/min helium flow, 25 μL/min ethanol flow, and 15–20 μL/min sample flow. We measured the jet diameter to be about 4.5 μm, with a flow rate of 40–45 μL/min under identical conditions to those used for the experiment. This translates into a jet speed of approximately 43 m/s[34]. During injection, sample was at room temperature, approximately 20 °C.

Rotation experiments (7Z2K, 7Z3U) were performed at PETRA-III beamline P11, delivering a 100 μm beam of 12 keV x-rays focused by a paired KB mirror system attenuated to 30% transmission[35]. Crystals were mounted robotically on a single-axis goniometer and held at 100 K using a cryojet (Oxford). During data collection, samples were rotated 200 degrees with frames read out from a DECTRIS Eiger detector at a distance of 200 mm every 0.2 degrees, for a total of 1000 images per crystal. Total dose per collection was approximately 1.05 MGy as determined by a calibrated diode measurement of x-ray flux ($0.7\cdot10^{12}$ ph/s at 100% transmission).

## Data analysis

During SFX experiments, online monitoring of the running experiment was performed with Karabo[31] v2 and OnDA[36] v22. The AGIPD geometry was refined against lysozyme data taken at the beginning and end of every shift using geoptimiser[37]. Preprocessing of images was performed with Cheetah[38] v2019.1 and subsequent crystallographic analysis was done with CrystFEL v0.9.1[39]. The MOSFLM algorithm was used for preliminary indexing[40], but all reported results used xgandalf[41]. Serial data merging was performed with partialator using the unity model. Data from rotation experiments with single crystals were processed with XDS[42] v10.01.2022. Prior to model building and refinement, XFEL datasets were resolution-truncated when CC* fell below 0.5[43]; rotation datasets employed CC1/2 > 0.5 as a cutoff. Following preliminary refinement, the resolution of 7PZQ was manually cut by an additional -0.1 Å, which improved the resulting refinement statistics and maps. All surface area calculations were performed with PISA[44] v1.48.

## Structure determination

Structures were determined by iterative rounds of model building in Coot[45] v0.9 and refinement with phenix.refine[46] v1.20.1, after molecular replacement with search models 6YNQ (for 7PXZ & 7Z3U), 7AKU (for 7PZQ) and 7AR5 (for 7Z2K). Disulfide, NOS, and SONOS bonds were enforced using bonding restraints generated by phenix. The rotameric states of the catalytic His41 were chosen based analyzing refined B factors of the sidechain in two possible conformations (supplementary Fig. S4, supplementary Table S2). All structures were refined with riding hydrogens.

## Analytical SEC

M[pro] was prepared in 20 mM Tris (pH 7.8) buffer supplemented with 150 mM NaCl and 1 mM EDTA, and for the indicated sample, 1 mM TCEP. The peroxide sample was produced by incubating protein with 1 mM hydrogen peroxide for 5 h prior to injection. Subsequently, protein solutions were spun down at 16,000 g for 5 min and applied to a Cytiva Superdex 75 10/300 increase column using a ÄKTA Pure system from Cytiva. Two peaks are observed in the resulting chromatograms at elution volumes consistent with dimer and monomer species. Concentrations were estimated by peak height at the position of peak maxima, and these were used to fit a two-state dimer dissociation model by least-squares regression. Raw data and processing scripts available upon request.

## Reporting summary

Further information on research design is available in the Nature Portfolio Reporting Summary linked to this article.

# Data availability

The data that support this study are available from the corresponding author upon request. Structural models, structure factor data, and associated metadata are available from the Protein Data Bank under PDB accession codes: 7PXZ (monoclinic XFEL reduced), 7PZQ (orthorhombic XFEL disulfide), 7Z2K (orthorhombic synchrotron reduced) & 7Z3U (orthorhombic synchrotron NOS/SONOS). Raw diffraction data are available from EuXFEL [https://doi.org/10.22003/XFEL.EU-DATA-002696-00]. Analytical SEC data and processing code

are available via Zenodo [https://doi.org/10.5281/zenodo.10616060]. The source data underlying Fig. 4 is provided as a Source Data file. Source data are provided with this paper.

## Code availability

The versions of Cheetah and CrystFEL used in this work are available from the respective websites: [https://www.desy.de/~barty/cheetah] and [https://www.desy.de/~twhite/crystfel].

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

## Acknowledgements

We acknowledge T. White for assistance during the XFEL experiment and C. Uetrecht for valuable discussions. T.J.L. was supported by a Helmholtz Young Investigator Award. H.B., C.B., S.B., H.N.C., T.J.L. acknowledge financial support obtained from the Cluster of Excellence 'Advanced Imaging of Matter' of the Deutsche Forschungsgemeinschaft (DFG) - EXC 2056 - project ID 390715994. In addition, H.B. and C.B. received support from BMBF via projects 05K19GU4 and 05K20GUB. The Helmholtz association funded S.F., J.L. and A.M. through the project FISCOV and P.Y.A.R. and A.M. through FragX. The Helmholtz association further supported the project by financing P.Y.A.R, J.L. and A.M. via the Impulse and Networking funds InternLabs-0011 'HIR3X'. W.E. was supported through the BMBF-funded project "ConScience" (project 16GW0277) and the Röntgen-Angstrom cluster project "X-ray drug design platform" (13K22CHB). D.T. was supported by the Slovenian Research Agency (ARRS; research program P1-0048, Infrastructural program IO-0048). L.R. was financed by BMBF project 13K18FLA. We acknowledge European XFEL in Schenefeld, Germany, for provision of x-ray free-electron laser beamtime at SPB/SFX as part of proposal 2696 and would like to thank the staff for their assistance. The authors are indebted to the XFEL Biology Infrastructure (XBI) for the provision of instrumentation that has enabled this experiment. Sample reservoirs and the anti-settling device employed in parts of the measurements presented here were designed and fabricated by the Max Planck Institute for Medical Research, Heidelberg, which also provided instruction in its use. This research was supported through computational resources (Maxwell cluster) and experimental facilities (PETRA-III beamline P11) operated by Deutsches Elektronen-Synchrotron DESY, Hamburg, Germany, a member of the Helmholtz Association HGF. We acknowledge the P11 staff for their invaluable help. All authors are indebted to Galen Correy, as well as an anonymous reviewer, for constructive criticism during the review process.

## Author contributions

Crystallization was performed by P.Y.A.R. and R.S. Size exclusion chromatography experiments were conducted by P.Y.A.R. Crystallographic refinement was performed by P.Y.A.R., R.S., D.O., D.T. and W.H. Authors M.G., A.R.M., S.G., A.C., A.R., B.C.S., B.N., C.K., C.S., F.H.M.K., A.T., W.E., G.E.P., G.M., H.K., H.B., H.H., J.Koliyadu, J.B., J.L., J.M., J.Knoška, K.L., L.B., M.S., M.K., M.V., P.V., P.M., R.d.W., R.B., R.L., S.H., S.F., T.G., T.S., V.S., Y.K., and O.M.Y contributed to the technical conduct of XFEL-based serial crystallography experiments. L.G., J.S., T.B., A.S.D., A.P.M., C.B., S.B., L.R., H.N.C., A.M., D.T., W.H. and T.J.L. conceived of the work, provided management oversight and technical comments on the manuscript. The manuscript, including figures and relevant data analysis, was drafted, and then revised, by P.Y.A.R., R.S., D.O., D.T., W.H. and T.J.L. The project was coordinated by T.J.L.

## Funding

## Competing interests

T.J.L. and A.S.D. are employees and shareholders of CHARM Therapeutics. T.G. is an employee of Sosei Heptares and A.S.D. was an employee of Sosei Heptares at the time the work was conducted. The remaining authors declare no competing interests.

## Additional information

**Patrick Y. A. Reinke** ⍟[1,19], **Robin Schubert** ⍟[2,19], **Dominik Oberthür** ⍟[1], **Marina Galchenkova**[1], **Aida Rahmani Mashhour** ⍟[1], **Sebastian Günther** ⍟[1], **Anaïs Chretien** ⍟[2], **Adam Round** ⍟[2], **Brandon Charles Seychell**[3], **Brenna Norton-Baker**[4,5], **Chan Kim** ⍟[2], **Christina Schmidt** ⍟[2], **Faisal H. M. Koua** ⍟[2], **Alexandra Tolstikova**[1], **Wiebke Ewert** ⍟[1], **Gisel Esperanza Peña Murillo** ⍟[1,6], **Grant Mills**[2], **Henry Kirkwood** ⍟[2], **Hévila Brognaro**[7], **Huijong Han**[2], **Jayanath Koliyadu** ⍟[2], **Joachim Schulz** ⍟[2], **Johan Bielecki** ⍟[2], **Julia Lieske** ⍟[1], **Julia Maracke**[1], **Juraj Knoska** ⍟[1,6], **Kristina Lorenzen**[2], **Lea Brings**[2], **Marcin Sikorski** ⍟[2], **Marco Kloos** ⍟[2], **Mohammad Vakili**[1,2], **Patrik Vagovic**[1,2], **Philipp Middendorf** ⍟[1], **Raphael de Wijn** ⍟[2], **Richard Bean** ⍟[2], **Romain Letrun** ⍟[2], **Seonghyun Han**[2,8], **Sven Falke** ⍟[1], **Tian Geng**[9], **Tokushi Sato**[2], **Vasundara Srinivasan** ⍟[7], **Yoonhee Kim**[2], **Oleksandr M. Yefanov**[1], **Luca Gelisio** ⍟[2], **Tobias Beck** ⍟[3,10], **Andrew S. Doré** ⍟[9,11], **Adrian P. Mancuso** ⍟[2,12,18],

**Christian Betzel** ⓘ [7,10], **Saša Bajt** ⓘ [1,10], **Lars Redecke** ⓘ [13,14], **Henry N. Chapman** ⓘ [1,6,10], **Alke Meents** ⓘ [1], **Dušan Turk** ⓘ [15,16], **Winfried Hinrichs** ⓘ [17] **& Thomas J. Lane** ⓘ [1,10,11] ✉

[1]Center for Free-Electron Laser Science CFEL, Deutsches Elektronen-Synchrotron DESY, Notkestr. 85, 22607 Hamburg, Germany. [2]European XFEL GmbH, Holzkoppel 4, 22869 Schenefeld, Germany. [3]Institute of Physical Chemistry, Department of Chemistry, Universität Hamburg, Grindelallee 117, 20146 Hamburg, Germany. [4]Max Plank Institute for the Structure and Dynamics of Matter, Luruper Chaussee 149, 22761 Hamburg, Germany. [5]Department of Chemistry, University of California at Irvine, Irvine, CA 92697–2025, USA. [6]Department of Physics, Universität Hamburg, Luruper Chaussee 149, 22761 Hamburg, Germany. [7]Institute of Biochemistry and Molecular Biology, Laboratory for Structural Biology of Infection and Inflammation, Department of Chemistry, Universität Hamburg, Build. 22a, c/o DESY, Notkestr. 85, 22607 Hamburg, Germany. [8]Gwangju Institute of Science and Technology, 123 Cheomdangwagi-ro, Buk-gu, Gwangju 61005, Republic of Korea. [9]Sosei Heptares, Steinmetz Building, Granta Park, Great Abington, CB21 6DG Cambridge, UK. [10]The Hamburg Centre for Ultrafast Imaging, Luruper Chaussee 149, 22761 Hamburg, Germany. [11]CHARM Therapeutics Ltd., B900 Babraham Research Campus, CB22 3AT Cambridge, UK. [12]La Trobe Institute for Molecular Science, Department of Chemistry and Physics, La Trobe University, Melbourne, VIC 3086, Australia. [13]Institute of Biochemistry, Universität zu Lübeck, Ratzeburger Allee 160, 23562 Lübeck, Germany. [14]Deutsches Elektronen-Synchrotron DESY, Notkestr. 85, 22607 Hamburg, Germany. [15]Jožef Stefan Institute, Jamova cesta 39, 1000 Ljubljana, Slovenia. [16]Centre of Excellence for Integrated Approaches in Chemistry and Biology of Proteins Jamova 39, 1000 Ljubljana, Slovenia. [17]Universität Greifswald, Institute of Biochemistry, Felix-Hausdorff-Str. 4, 17489 Greifswald, Germany. [18]Present address: Diamond Light Source, Harwell Science and Innovation Campus, OX11 0DE Didcot, UK. [19]These authors contributed equally: Patrick Y. A. Reinke, Robin Schubert. ✉e-mail: thomas.lane@desy.de

