## [Peer Review File · Nature Communications]

SARS-CoV-2 Mpro responds to oxidation by forming disulfide and NOS/SONOS bondsReviewer #1 (Remarks to the Author):

The manuscript by Reinke et al. investigates the structural basis for the change in activity and oligomeric state of Mpro from SARS-CoV-2 upon oxidation. Mpro is active as a homodimer, oxidation has been shown to shift the oligomeric state to monomeric (with loss of activity), while reduction restores the dimer and activity. Previously reported mutagenesis of Mpro's cysteines suggests that a disulfide bond between the catalytic cysteine (C145) and a distal cysteine (C117) is involved in the shift in activity and oligomeric state. Reinke et al. report a crystal structure of the oxidized form of Mpro in the P212121 space group determined using XFEL radiation and serial crystallography that contains a disulfide bond between C117 and C145. Structures of the reduced protein in the C2 space group using XFEL radiation and of the reduced protein in the P212121 space group are reported, along with a co-crystal structure of Mpro with a ligand covalently bound to C145 in the P212121 space group. The major success of this paper is in helping to understand the structural basis for the redox regulation of Mpro. The weaknesses of this paper are in the failure to discuss a previously reported structure of oxidized Mpro (Major point 1) and the lack of clarity around how different crystals were grown (major point 3, minor points 2 & 3). This paper adds to the growing literature describing the redox control of Mpro and will be of interest to the structural biology community.

Major points

1. An X-ray crystal structure of SARS-CoV-2 Mpro with the Cys117-Cys145 disulfide has been reported for the H163A variant (<https://www.biorxiv.org/content/10.1101/2022.12.16.520794v1>, PMID 37699927, PDB 8DDL). Please cite this work and include a brief discussion of the similarities / differences with the oxidized structure reported here (PDB 7PZQ). This seems particularly important given that the same disruption of the dimer interface shown in Figure 3 is seen in the 8DDL structure.
2. The manuscript would be strengthened by including more support for the SONOS/NOS bridges in the 7Z3U structure. The unit cells for 7Z2K and 7Z3U seem isomorphous, therefore an FO-FO map using phases calculated using the 7Z2K structure might be appropriate. Otherwise, an FO-FC map calculated prior to modeling the linkage could be shown. Supplementary figure would be fine. This seems particularly important for the K102/C156 NOS bond, given that this hasn't been reported previously and the density is comparatively weak.
3. Please clarify in the text how the crystals were grown for the XFEL experiments. I can't work out if: A) two separate batch crystallization experiments were set up with protein purified either with or without TCEP using the same crystal seeds, and this led to two batches containing reduced or oxidized crystals, and these were then used for two separate XFEL experiments or, B) one batch crystallization experiment was set up using protein purified in the absence of TCEP, upon indexing the diffraction images it was noticed that two crystal forms were present. Structures determined with merged data from the two crystal forms showed one to be oxidized and one to be reduced. The graphic in Figure 1 and some of the text seems to suggest option B, but the first sentence of the results (page 3 line 147) suggests option A. Perhaps update Figure 1A to clarify this (also minor points 1 and 2).

Minor points

1. Please include the crystallization conditions (and whether a reducing agent was used) for the initial Mpro crystals used to create seeds (e.g. page 17 line 429).
2. It's unclear to me how crystals of the reduced form in the P212121 space group were obtained. The results state that seeds prepared from the oxidized form were used (page 5, line 164). Were the crystals selected for crushing/seeding based on morphology from crystallization drops containing both crystal types? If this is the case, then presumably a structure of the oxidized form using synchrotron radiation could be obtained? This information would be helpful for others trying to reproduce your crystals.
3. How was the resolution cut-off decided for the X-ray data for structures reported in this work? Table 1 does not list CC1/2, and the I/ σ I is very low in the case of 7PZQ (0.04 in the highest resolution shell). Please add details to the Methods.
4. The B-factors for protein atoms are (slightly) lower for the 2.25 Å structure (7PZQ) compared to the 1.75 Å structure (7PXZ). The water B-factors are substantially lower (55 vs 41 Å²). Can the authors

comment on this?

5. The reduced Mpro structure determined by XFEL radiation (PDB 7Z3U) has hydrogens added to the water molecules. How were the hydrogens refined? E.g., riding model? Please add details to the Methods.
6. The oxidized Mpro structure determined by XFEL radiation (PDB 7PZQ) was not refined with hydrogens. Was this because it is lower resolution compared to the others?
7. Why were different protein concentrations used in the analytical SEC experiment for samples with different treatments? There seems to be some data points missing (Figure 4) - were these excluded? Some explanation in the figure legend or methods would be helpful.
8. Please include the size-exclusion chromatograms used to generate Figure 4.
9. The colors for the sticks in Figure 3A are hard to distinguish under the colored surface. Perhaps make the surface more transparent?
10. Please check that the catalytic histidine (His41) is correctly modeled across the seven structures / protomers. Although distinguishing based on electron density at this resolution is not possible, I'm guessing they should be the same rotamer based on hydrogen bonding patterns.
11. The methods say that PDB 7AR6 was used for molecular replacement, PDB metadata says 7AKU for 7PXZ and 7PZQ and 6YNQ for 7Z3U.
12. Please include the PDB codes in figure legends when presenting structures.
13. Consider uploading diffraction images (for synchrotron datasets) to proteindiffraction.org or similar. Removes a barrier for access.
14. Why was EDTA included in the buffer when trying to oxidize the protein (page 17, line 426)? My understanding is that EDTA may chelate metal ions and reduce the rate of oxidation?
15. The intro/results could be modified to make it clear that you have reported the co-crystal structure with S-calpeptin previously, and that the present work analyzes part of these structures (the SONOS/NOS bonds).

Happy to review non-anonymously,
Galen Correy

Reviewer #2 (Remarks to the Author):

The authors report structural studies on SARS-CoV-2 Mpro, an important drug target for fighting COVID-19. Previous reports had indicated that Mpro might be subject to redox regulation in the form of a disulfide-dithiol switch (C117-C145) and lysine-cysteine linkages involving C22, C44 and K61 (NOS and SONOS). The disulfide switch had escaped a structural analysis so far, while there have been some indications for the NOS and SONOS bridges as discussed by the authors. Here, the authors use the XFEL technology to systematically analyze redox modifications of SARS-CoV-2 under controlled redox conditions. Most remarkably, they could solve a structure of Mpro with the proposed C117-C145 disulfide bond and carefully describe not only the protocol to obtain well suited crystals but also the structural consequences with respect to the dimer interface and stability of the dimer (only dimer is active). Remarkably, they further observe a variety of lysine-cysteine linkages including C22-K61 (NOS), C22-K61-C44 (SONOS) and C156-K102 (NOS). Overall, the study is technically extremely well executed and provides very important insights into redox modifications and a potential redox regulation of SARS-CoV-2. The existing literature on this topic is well reflected in the discussion.

Reviewer #1 (Remarks to the Author):

The manuscript by Reinke et al. investigates the structural basis for the change in activity and oligomeric state of Mpro from SARS-CoV-2 upon oxidation. Mpro is active as a homodimer, oxidation has been shown to shift the oligomeric state to monomeric (with loss of activity), while reduction restores the dimer and activity. Previously reported mutagenesis of Mpro's cysteines suggests that a disulfide bond between the catalytic cysteine (C145) and a distal cysteine (C117) is involved in the shift in activity and oligomeric state. Reinke et al. report a crystal structure of the oxidized form of Mpro in the P212121 space group determined using XFEL radiation and serial crystallography that contains a disulfide bond between C117 and C145. Structures of the reduced protein in the C2 space group using XFEL radiation and of the reduced protein in the P212121 space group are reported, along with a co-crystal structure of Mpro with a ligand covalently bound to C145 in the P212121 space group. The major success of this paper is in helping to understand the structural basis for the redox regulation of Mpro. The weaknesses of this paper are in the failure to discuss a previously reported structure of oxidized Mpro (Major point 1) and the lack of clarity around how different crystals were grown (major point 3, minor points 2 & 3). This paper adds to the growing literature describing the redox control of Mpro and will be of interest to the structural biology community.

We want to thank the reviewer up for diving into the details of our work and for providing actionable suggestions for improvement. We respond point-by-point below and document the changes made during revision.

Major points

1. An X-ray crystal structure of SARS-CoV-2 Mpro with the Cys117-Cys145 disulfide has been reported for the H163A variant. Please cite this work and include a brief discussion of the similarities / differences with the oxidized structure reported here (PDB 7PZQ). This seems particularly important given that the same disruption of the dimer interface shown in Figure 3 is seen in the 8DDL structure.

We agree with the reviewer. This work on the H163A mutant only came to our attention following its publication in *Nature Communications*, immediately after we submitted our paper for review. The work is of course relevant for our own results and – as the reviewer says – deserves acknowledgement and discussion.

In the relevant paper, Tran *et al.* report the observation of a disulfide between Cys117-Cys145 in M^{pro} H163A. They compared the structural changes in the H163A mutant to wild type M^{pro} (PDB 7BB2). The changes they report upon disulfide formation generally agree with our findings. One notable difference is that we do not observe a complete outward flip of the loop spanning residues S139-S147, containing F140, that was observed in the H163A mutant.

We discuss this in more detail in our revised manuscript. In the Introduction, we acknowledge the work of Tran et al (pg. 3 line 134),

“Supporting this, Tran et al. recently reported the structure of an oxidized mutant of M^{pro}, the H163A variant, that contained a disulfide bond linking C117 to C145.¹⁴ The

H163A modification inactivates the enzyme, even under reducing conditions. Removal of the H163 sidechain disrupts a π -stacking interaction with F140, which is part of the loop that forms the transition state-stabilizing oxyanion hole. Disruption of the oxyanion hole was observed in structures of both the oxidized (C117-C145) and reduced (no disulfide) enzyme, explaining the loss in activity. However, it was unclear how the C117-C145 disulfide in the H163A variant related to the behavior of the wild type.”

Then, in our Results section, we briefly mention the remodeling of the loop Tran *et al.* discuss (pg. 9, line 243),

“Moreover, the position of the loop containing the active cysteine, spanning S139 to S147, rearranges upon disulfide formation (Fig. S2).”

Which we discuss elaborate on in the Discussion section (pg. 15 line 412),

“By studying the H163A M^{pro} mutant, Tran and colleagues were able to crystallographically observe the same C117-C145 disulfide bond we found in wild-type M^{pro}¹⁴. In that work, they observed that the loss of the imidazole sidechain in H163 removed a favorable π -stacking interaction with F140, resulting in displacement of a mobile loop containing F140 (Fig. S2). Further, this loop motion displaces the N and C termini, resulting in dimer interface disruption. Importantly, the loop containing F140 forms the oxyanion hole, which stabilizes the transition state during proteolysis. In our disulfide-containing structure, the F140 loop does not reach the same extreme position as observed by Tran *et al.* but is disrupted from its conformation in the corresponding reduced structures (Fig. S2). Our work therefore supports the hypothesis of Tran *et al.*, that C117-C145 can form in a similar fashion in both the wild type and H163A enzymes but demonstrates that the extensive remodeling of the F140 loop in the H163A structure is particular to that variant.”

This discussion is complemented by a new supplemental figure (S2) that shows the F140 loop for our structures overlaid with that from the H163A mutant from Tran *et al.*

2. The manuscript would be strengthened by including more support for the SONOS/NOS bridges in the 7Z3U structure. The unit cells for 7Z2K and 7Z3U seem isomorphous, therefore an FO-FO map using phases calculated using the 7Z2K structure might be appropriate. Otherwise, an FO-FC map calculated prior to modeling the linkage could be shown. Supplementary figure would be fine. This seems particularly important for the K102/C156 NOS bond, given that this hasn't been reported previously and the density is comparatively weak.

This was a good suggestion, and we've added a supplemental figure (Fig. S3) that shows the original $2mF_o-DF_c$ maps, (composite) OMIT maps, and finally an isomorphous difference map (7Z3U-7Z2K, with phases from 7Z2K), each for both protomers in the ASU. In the caption (SI pg. 4), we conclude:

“While there is some density between K102 and C156 for chain A, it is not possible to confidently conclude a NOS bond exists. In contrast, for chain B, all three map types show evidence of a NOS linkage at partial occupancy.

3. Please clarify in the text how the crystals were grown for the XFEL experiments. I can't work out if: A) two separate batch crystallization experiments were set up with protein purified either with or without TCEP using the same crystal seeds, and this led to two batches containing reduced or oxidized crystals, and these were then used for two separate XFEL experiments or, B) one batch crystallization experiment was set up using protein purified in the absence of TCEP, upon indexing the diffraction images it was noticed that two crystal forms were present. Structures determined with merged data from the two crystal forms showed one to be oxidized and one to be reduced. The graphic in Figure 1 and some of the text seems to suggest option B, but the first sentence of the results (page 3 line 147) suggests option A. Perhaps update Figure 1A to clarify this (also minor points 1 and 2).

Sorry about this confusion – the reviewer's “option A” is correct, we set up two separate batch crystallization experiments to generate the reduced and oxidized crystals as part of a larger XFEL experiment. We only realized after the reviewer's comment that we should carefully clarify that the crystals were not produced in the *same* batch, which can produce multiple lattices for some systems.

We've taken the following steps to ensure the revised manuscript is clear on this point:

1. Updated the text at the beginning of the results section (pg. 3, line 156) to read:

“The crystals merged to determine these two structures were grown in two separate batch crystallization experiments. In the first, the resulting structure was fully reduced due to the presence of 1 mM TCEP during purification. In the second experiment, TCEP was omitted, and Mpro exposed to air over time spontaneously crystallized into a different space group, despite being crystallized under otherwise the same conditions (buffer, temperature).”

2. Elaborated in the caption of Figure 1, as suggested by the reviewer (pg. 5, line 187), so that it now reads:

“In a separate batch crystallization experiment, after oxidation by air...”

3. Finally, in the methods, we've written (pg. 19, line 500):

“The reduced and oxidized forms were crystallized separately.”

Minor points

1. Please include the crystallization conditions (and whether a reducing agent was used) for the initial Mpro crystals used to create seeds (e.g. page 17 line 429).

Thanks for highlighting this omission in our protocol. We've added the following detail:

“Seed crystals were grown from protein purified in the presence of 0.5 mM TCEP, using a sitting drop geometry by combining 250 nL Mpro protein solution (6.25 mg/ml) and 250 nL precipitant (25 % PEG1500, 0.1 M MIB buffer pH 7.5, 5% DMSO), as reported previously³⁰.”

On page 19, line 492.

2. It's unclear to me how crystals of the reduced form in the P212121 space group were obtained. The results state that seeds prepared from the oxidized form were used (page 5, line 164). Were the crystals selected for crushing/seeding based on morphology from crystallization drops containing both crystal types? If this is the case, then presumably a structure of the oxidized form using synchrotron radiation could be obtained? This information would be helpful for others trying to reproduce your crystals.

Hopefully the response to Major Point 3 above makes it clear that we obtained a pure sample of oxidized crystals in batch.

We've tried many times to obtain a structure of the C117-C145 bond using synchrotron radiation. Some structures of M^{pro} we've collected as part of work do show weak residual density at the position occupied by the disulfide, but if the linkage was present, it was at very low occupancy and impossible to model confidently. Hence, we cannot provide a protocol that we feel will confidently lead another research team to obtain those results using synchrotron radiation.

At the same time, we can't claim it's impossible, and that XFEL radiation is necessary to obtain data of the quality we report in our current manuscript. We can only state that the XFEL approach we took is sufficient, which we make clear in the manuscript. To emphasize this point, we've expanded the Discussion on this topic, writing on pg. 15 starting line 390:

“While we have been able to grow large single crystals from air-oxidized protein in the orthorhombic space group (Fig. S1) and have studied these with synchrotron radiation, we have not observed clear C117-C145 disulfide density in those studies. In contrast, diffraction of XFEL light from microcrystals clearly shows the C117-C145 disulfide. We considered the hypothesis that XFEL radiation may have allowed us to observe this modification via “radiation damage free” data collection, as the x-ray exposure (<100 fs) is much more rapid than the nuclear motions required for the two cysteine sidechains to adopt significantly different positions following x-ray induced reduction^{25–27}. In this model, structures obtained at the synchrotron would appear fully reduced, because during data collection at the synchrotron, x-ray induced cleavage of the disulfide would reverse the structural changes that accompany C117-C145 crosslinking. This would require an implausibly large rearrangement of the structure under cryogenic conditions. The success of Tran et al. in capturing this disulfide with synchrotron radiation in the H163A variant¹⁴ is further evidence against this model. Their ability to obtain a structure clearly showing the C117-C145 disulfide suggests that this modification is not uniquely sensitive to reduction by x-rays.

Alternatively, it may be that the large crystals containing significant C117-C145 population are less well ordered than their microcrystalline counterparts, and XFEL radiation allowed us to study such microcrystals. Our data only show that XFEL radiation is sufficient to observe the C117-C145 disulfide in wild type Mpro, but we cannot conclude that such radiation is necessary to preserve the C117-C145 modification.”

To aid reproducibility, we’ve included micrographs of large crystals grown in both space groups in our new supplement (Fig. S1).

3. How was the resolution cut-off decided for the X-ray data for structures reported in this work? Table 1 does not list CC1/2, and the I/σ is very low in the case of 7PZQ (0.04 in the highest resolution shell). Please add details to the Methods.

When the reviewer pointed out a I/σ value of 0.04 in the highest resolution shell, we realized we had made some kind of typographical or copy/paste error. The correct value (as reported in the PDB) should be 1.07, which we’ve corrected in the text. Thanks to the reviewer for their sharp eye!

Additionally, we’ve added information about how we truncated the data, writing in the methods (pg. 20, line 532):

“Prior to model building and refinement, XFEL datasets were resolution-truncated when CC^* fell below 0.5; rotation datasets employed $CC1/2 > 0.5$ as a cutoff. Following preliminary refinement, the resolution of 7PZQ was manually cut by an additional ~ 0.1 Å, which improved the resulting refinement statistics and maps.”

We originally filled out the Nature Table 1 template, which did not include $CC1/2$ and CC^* , but have added these values in the revised manuscript. Both the serial XFEL data and rotation data were cut using rules of thumb in common practice, even though they differ slightly.

4. The B-factors for protein atoms are (slightly) lower for the 2.25 Å structure (7PZQ) compared to the 1.75 Å structure (7PXZ). The water B-factors are substantially lower (55 vs 41 Å²). Can the authors comment on this?

It’s an interesting point, and one that our team discussed. The protein B factor trend is as expected. At this time, however, we are unable to provide a conclusive answer as to why the water B factors differ. The packing and solvent content of the two crystal forms, and consequentially where ordered waters appear, are not the same in the two structures. We can’t discount the fact that waters are added and removed based in part on manual decision making. Since we can’t conclude anything definitively with the data we have in hand, we decided to refrain from adding any discussion on the details of B factors to the text.

Note that we’ve re-refined the structures (see minor point 5, immediately below), so the B factors have changed slightly. The trend the reviewer pointed out remains; the water B factors of 7PZQ are still lower than those for 7PXZ.

5. The reduced Mpro structure determined by XFEL radiation (PDB 7Z3U) has hydrogens added to the water molecules. How were the hydrogens refined? E.g., riding model? Please add details to the Methods.

Thanks for pointing out this omission. The hydrogens originally come from model building and refinement performed in *MAIN* [see Acta Cryst. D (2013), 10.1107/ S0907444913008408], written and maintained by one of the authors (DT), which protonates waters as a standard practice. Subsequent model building and refinement retained these riding water hydrogens.

Given the reviewer's comments and apparent confusion here, our team decided that simplifying and unifying the refinement/deposition of all datasets, especially a uniform treatment of hydrogens, would enable maximum comparability and minimum confusion for future readers. Therefore, we've removed the hydrogens from these waters, added riding hydrogens to all protein portions of the models, re-performed final refinement in *phenix*, and deposited revised structures in the PDB.

To make the new procedure clear to readers, we've added the following expanded protocol to the Methods section (pg. 20 line 537),

“Structure determination. Structures were determined by iterative rounds of model building in Coot47 and refinement with phenix.refine⁴⁸, after molecular replacement with search models 6YNQ (for 7PXZ & 7Z3U), 7AKU (for 7PZQ) and 7AR5 (for 7Z2K). Disulfide, NOS, and SONOS bonds were enforced using bonding restraints generated by phenix. The rotameric states of the catalytic His41 were chosen based analyzing refined B factors of the sidechain in two possible conformations (Fig S4, Table S4). All structures were refined with riding hydrogens.”

6. The oxidized Mpro structure determined by XFEL radiation (PDB 7PZQ) was not refined with hydrogens. Was this because it is lower resolution compared to the others?

Please see our response to point 5 above.

7. Why were different protein concentrations used in the analytical SEC experiment for samples with different treatments? There seems to be some data points missing (Figure 4) - were these excluded? Some explanation in the figure legend or methods would be helpful.

Thanks for pointing this out, we see the reviewer's point of view. We did not exclude any data points. We did, however, correct for the concentration of protein in each SEC run using the absorbance at 280 nm of each sample. In some cases, this caused a large shift of the position of these datapoints along the x-axis, and made it appear like some data might have been omitted. Further, we did not collect precisely the same concentrations for every condition, but originally only aimed to span the K_D .

Because these combined factors made the presentation of the data less clean than we would like, we elected to re-perform the experiment during review. Because of the time that elapsed, the SEC setup we employed was different (tubing, column). The resulting data are modestly

improved but differ in some expected ways from the old data (*e.g.*, where the peaks elute). Therefore, in the revised manuscript, we present only the new data.

Interestingly, these improvements allowed us to observe a shift in the elution volume of the oxidized monomer species. We write (pg. 11, line 295):

“Further, upon expose to peroxide, the observed monomer peak elutes notably earlier, at approximately 0.49 column volumes vs. 0.47, suggesting oxidation has produced a less compact species.”

Further, we would like to note for the reviewer’s sake that previously we employed 5 mM peroxide as part of our SEC experiments. In repeating the experiments with a fresh stock of peroxide, however, we discovered this completely degraded our protein. Titration of the old stock with potassium permanganate confirmed a loss of peroxide concentration due to the age of the stock. We are glad to have caught this prior to publication.

Therefore, we reduced the concentration to 1 mM (confirmed by permanganate titration), which gave qualitatively similar results to our previous experiment, but with a lower K_D (19 +/- 14.8 μ M now vs. 97 +/- 43 μ M previously). This may be because the effective concentration of peroxide is lower, because of our improved SEC setup, or both.

Finally, we removed the claim that there is a significant difference between protein purified without TCEP and with 1 mM TCEP. We now believe this is a function of how fresh the protein stock is, as the protein appears to slowly oxidize in air over time. Interestingly, this seems to be consistent with the report of Funk et al., who state that even peroxide-driven oxidation takes hours to equilibrate (see 10.1038/s41467-023-44621-0, pg. 3).

We report results for freshly purified protein in our revision, and state this clearly in the manuscript (“Freshly purified protein, exposed to air for only a few hours, ...”, pg. 11, line 290) which is a reproducible procedure. Exposure to air can shift this equilibrium over time. Because we have not studied this in depth, we report the data for freshly prepared protein, which is reproducible. This does not change our conclusions, which are focused on a comparison of the fully reduced (TCEP) and more aggressively oxidized (peroxide) samples.

These data are now reported in the manuscript, alongside the raw SEC traces in Fig. 4, panel A (see response to point 8, below).

8. Please include the size-exclusion chromatograms used to generate Figure 4.

We’ve drawn these into figure 4 and archived the raw data and processing code to Zenodo.

9. The colors for the sticks in Figure 3A are hard to distinguish under the colored surface. Perhaps make the surface more transparent?

Good point, we appreciate the opinion. We’ve made the surface more transparent and removed the red/blue charge colors on the left-hand surface, allowing the sticks and their colors to come into the foreground.

10. Please check that the catalytic histidine (His41) is correctly modeled across the seven structures / protomers. Although distinguishing based on electron density at this resolution is not possible, I'm guessing they should be the same rotamer based on hydrogen bonding patterns.

The reviewer's comment on this topic sparked some interesting discussion amongst our team. We start by noting that the rotameric state of His41 does not have implications for our scientific conclusions, but that nonetheless we attempted to model this important residue as well as possible in case our structures are used by others in the future.

As the reviewer suspected, it is challenging to distinguish the two rotamers based on the data we have available. However, we adopted the following procedure based on the idea that the refined B factors of all atoms in the His41 side chain should be comparable, especially the fact that for a stable refinement, the B factor of NE2 should not be much higher than CE1. First, we defined the two rotamers (supplement pg. 5, caption of Fig. S4),

“the *syn* conformation (cyan), histidine N_δ is proximal to a conserved and strongly bound water in the active site. The *anti* conformer (yellow) is produced by a 180° flip of the sidechain around the C_β-C_γ bond. In both cases the catalytic water exhibits hydrogen bonds to His164(ND1), Asp187(OD2), and His41(N)“

Then, for each instance of His41 (7 unique chains across 4 structures), we performed a refinement of atomic positions and B factors for both the *syn* and *anti* rotamers. Key parameters are reported in Table S1 (supplement pg. 6). In the table caption, we report,

“Total B factors for these two conformers are very close, suggesting our data do not significantly favor one conformer over the other, and that both may be populated in superposition. The distances between H41 N_ε and C145 S_γ are not substantially different in one conformer vs. the other. Thus, both conformers are within the necessary hydrogen bonding distance between imidazolium 41 and thiolate 145 to produce the initial state of the catalytic cycle. In the absence of other information, we chose the *syn* or *anti* conformation based on the individual B factors of corresponding imidazole C/N-atom pairs, e.g. the B factor of NE2 should not be much higher than CE1. The final modeled rotameric state is indicated with bold font. Note that in 7PZQ and 7Z3U that Cys145 is in a disulfide and thiohemiacetal form, respectively, and therefore not capable of catalysis.”

Table S1 indicates the final rotameric state of each His41, which have been deposited as part of our PDB update.

11. The methods say that PDB 7AR6 was used for molecular replacement, PDB metadata says 7AKU for 7PXZ and 7PZQ and 6YNQ for 7Z3U.

Thanks for spotting this error, which was caused by miscommunication on our team. The PDB is correct, and we've updated the manuscript to address this (pg. 20, line 538):

“... after molecular replacement with search models 6YNQ (for 7PXZ & 7Z3U), 7AKU (for 7PZQ) and 7AR5 (for 7Z2K)”

12. Please include the PDB codes in figure legends when presenting structures.

Good suggestion, done.

13. Consider uploading diffraction images (for synchrotron datasets) to proteindiffraction.org or similar. Removes a barrier for access.

Our priority so far has been to address the issues raised by the reviewer; our team is looking into archiving the raw image data for the synchrotron datasets as well as the XFEL data. Both are currently stored on tape, so this will take some time, and we do not wish to delay publication while we sort this out. If the editor agrees, our preference is to proceed with the paper's evaluation and revisit adding a reference to the raw data immediately prior to publication.

We note that all the data requested for publication by *Nature Communications* is already publicly archived or ready to be released: structure factors are available *via* the PDB and our raw SEC data and code will be released on Zenodo/Github if our manuscript is accepted.

14. Why was EDTA included in the buffer when trying to oxidize the protein (page 17, line 426)? My understanding is that EDTA may chelate metal ions and reduce the rate of oxidation?

This might have been a small miscommunication. EDTA was included in the final SEC buffer for both oxidized and reduced samples. In the methods section we write (pg. 19 line 488),

“a final size exclusion chromatography was performed with an S200 Superdex column using 20 mM TRIS, pH 7.8, 150 mM NaCl, 1 mM TCEP and 1 mM EDTA, while for the oxidized form TCEP was omitted.”

This was used previously in Ref. 2, which in turn adopted a protocol from Hilgenfeld's group, see Zhang *et al.*, *Science* 368, 409-412 (2020).

15. The intro/results could be modified to make it clear that you have reported the co-crystal structure with S-calpeptin previously, and that the present work analyzes part of these structures (the SONOS/NOS bonds).

Good suggestion to make this more explicit. We now spell this out, writing (pg. 6, line 199),

“The resulting structure was previously reported in a paper describing the ligand binding pose²⁰. Unexpectedly, the same structure exhibits a rich pattern of oxidative modifications, which we discuss here.”

Happy to review non-anonymously,
Galen Correy

We are thankful – sincerely – for such a thorough review, which has made our paper significantly better!

Reviewer #2 (Remarks to the Author):

The authors report structural studies on SARS-CoV-2 Mpro, an important drug target for fighting COVID-19. Previous reports had indicated that Mpro might be subject to redox regulation in the form of a disulfide-dithiol switch (C117-C145) and lysine-cysteine linkages involving C22, C44 and K61 (NOS and SONOS). The disulfide switch had escaped a structural analysis so far, while there have been some indications for the NOS and SONOS bridges as discussed by the authors.

Here, the authors use the XFEL technology to systematically analyze redox modifications of SARS-CoV-2 under controlled redox conditions. Most remarkably, they could solve a structure of Mpro with the proposed C117-C145 disulfide bond and carefully describe not only the protocol to obtain well suited crystals but also the structural consequences with respect to the dimer interface and stability of the dimer (only dimer is active). Remarkably, they further observe a variety of lysine-cysteine linkages including C22-K61 (NOS), C22-K61-C44 (SONOS) and C156-K102 (NOS).

Overall, the study is technically extremely well executed and provides very important insights into redox modifications and a potential redox regulation of SARS-CoV-2. The existing literature on this topic is well reflected in the discussion.

We thank the reviewer for the time spent reading and analyzing our work. Our team appreciates the generous comments about our paper and has striven to improve it during the review process.

Reviewer #1 (Remarks to the Author):

The revised manuscript addresses all my concerns. I appreciate the time and care that the authors took in responding to my comments and congratulate them on the revised manuscript.

Reviewer #2 (Remarks to the Author):

The authors have carefully revised the ms in light of the reviewers' comments such that acceptance is supported by this referee.